# A Decision-Making Strategy for Car Following Based on Naturalist Driving Data via Deep Reinforcement Learning

**DOI:** 10.3390/s22208055

**Published:** 2022-10-21

**Authors:** Wenli Li, Yousong Zhang, Xiaohui Shi, Fanke Qiu

**Affiliations:** Key Laboratory of Advanced Manufacture Technology for Automobile Parts, Ministry of Education, Chongqing University of Technology, Chongqing 400054, China

**Keywords:** deep reinforcement learning, naturalist driving data, speed-acceleration distribution, action’s varying constraint

## Abstract

To improve the satisfaction and acceptance of automatic driving, we propose a deep reinforcement learning (DRL)-based autonomous car-following (CF) decision-making strategy using naturalist driving data (NDD). This study examines the traits of CF behavior using 1341 pairs of CF events taken from the Next Generation Simulation (NGSIM) data. Furthermore, in order to improve the random exploration of the agent’s action, the dynamic characteristics of the speed-acceleration distribution are established in accordance with NDD. The action’s varying constraints are achieved via a normal distribution 3σ boundary point-to-fit curve. A multiobjective reward function is designed considering safety, efficiency, and comfort, according to the time headway (THW) probability density distribution. The introduction of a penalty reward in mechanical energy allows the agent to internalize negative experiences. Next, a model of agent-environment interaction for CF decision-making control is built using the deep deterministic policy gradient (DDPG) method, which can explore complicated environments. Finally, extensive simulation experiments validate the effectiveness and accuracy of our proposal, and the driving strategy is learned through real-world driving data, which is better than human data.

## 1. Introduction

With rapid growth in the scale of urban traffic and the standing increment of vehicles, car following (CF) has become the most common driving behavior in daily driving. It has been widely used in microscopic traffic simulation and autonomous driving [1]. For autonomous vehicles (AVs), safe and comfortable driving will increase passenger satisfaction and trust, minimize fuel consumption, and benefit auto owners financially. Poor CF performance will lead to traffic congestion and oscillation [2].

The research object for car-following behavior is the interaction between people and vehicles. It describes the interaction mechanism of vehicles on the road in the process of longitudinal movement, taking into account factors such as safety, comfort, and efficiency [3,4]. The driver models related to CF models are generally established based on two approaches: the rule-based approach and the supervised learning approach [5,6,7]. The former relies primarily on a differential equation model to develop a CF strategy, ranging from simple control logic to advanced control logic, such as proportion integral differential (PID) control [8,9], fuzzy control [10], and model predictive control (MPC) [11,12]. Due to the model’s restrictions, a CF strategy based on a differential equation model lacks the ability to generalize to unknown situations in a real traffic environment, and researchers are unable to enumerate every scenario that might arise during the CF process. The latter typically relies on data provided by human demonstrations, imitating driving strategies from data extracted from human driving in a supervised manner, such as deep neural networks (DNNs), long short-term memory (LSTM), and k-nearest neighbor (KNN) [13,14,15]. However, it is challenging to develop an ideal driving strategy because it relies heavily on a large amount of annotated driving data that essentially only simulates human driving behavior rather than optimizing for safety, efficiency, and comfort. Furthermore, it is insufficient to derive only empirical control rules from natural driving data. The advantages of optimal control should be exploited and thoroughly combined with the driver’s driving characteristics.

Aiming to address this limitation, the application of deep reinforcement learning (DRL) methods to the processing of vehicle decision control has attracted the widespread attention of researchers [16]. DRL combines deep learning and reinforcement learning to deal with high-dimensional state space and discrete or continuous action spaces in decision-making problems, enabling agents to make autonomous decisions in complex scenarios [17]. Accordingly, DRL-based methods can be applied at many levels, such as robots [18], traffic lights [19], autonomous vehicles [6], and hybrid vehicle energy management [20]. Using real-world driving data to drive model training can achieve better control performance; that is, exploiting expert knowledge can provide the best training samples or preferences for the agent to guide action exploration in the training process, thereby improving their learning and adaptability.

In light of existing works, we use the characteristics of naturalist CF behavior, combined with DRL and expert knowledge, to form an adaptive learning method. The main contributions of this paper are as follows: (1) We analyze the dynamic characteristics of CF from NGSIM data, and frequency statistics characterize the distribution of each characteristic parameter in the CF process. The correlation coefficient is used to analyze the significance of each characteristic parameter to following vehicle (FV) speed. (2) We propose to utilize the deep deterministic policy gradient (DDPG) algorithm for decision-making to obtain an autonomous CF control strategy for AVs. In addition, utilizing naturalist driving data (NDD) to establish the dynamic characteristics of speed-acceleration distribution enhances random exploration of the agent’s actions; to realize the action-varying constraints, it fits the relationship curve according to the normal distribution of the 3σ boundary points. A multiobjective reward function is designed that takes into account CF behavior traits as well as driving efficiency, comfort, and safety objectives. (3) Validation: The developed strategy is evaluated through extensive simulations. The simulation results validate the strategy’s effectiveness in learning and evaluating the safety, efficiency, and comfort performance of the CF decision-making process.

The paper is organized as follows: The related work is introduced in Section 2. Section 3 analyzes the characteristics of car-following behavior based on naturalist driving data. Then, Section 4 presents details of the car-following decision-making strategy model based on deep reinforcement learning. Extensive simulations are discussed in Section 5, and conclusions are summarized in Section 6.

## 2. Related Work

In the past few decades, researchers have proposed many CF optimal control algorithms and strategies [21]. For traditional car-following strategies based on differential equation models, fuzzy self-optimizing PID [9] and fuzzy logic [10] have been proposed to adapt to nonlinear and time-varying traffic flow behavior. In order to improve driving comfort and robustness, Schmied et al. [22] proposed a CF method under multilane traffic conditions. However, the difficulty in developing and applying adaptive cruise control (ACC) lies in establishing a multiobjective control strategy that includes safety, comfort, and economy. Among them, the ability of MPC to handle multiple constraints by rolling the horizon has been widely used to solve the problem of CF. Goni-Ros et al. [11] established an MPC car-following model based on a constant time headway (THW). With the rapid development of data-driven technology, Moon et al. [8] proposed a PID-controlled car-following model considering human factors based on a large amount of real test data. Bolduc et al. [12] proposed an integrated, optimal ACC driver multimodel, which uses MPC to track the reference trajectory that best represents the driver’s style model. However, the real traffic environment is full of complexity and randomness, which limits the flexibility and generalization of traditional control methods. Likewise, researchers cannot enumerate all the situations that may occur in the process of car following.

With the rapid development of communication technology, the application of communication technology to the CF model has become a research hotspot, making multiple CFs connected into a platoon, which expands the vehicle’s perception ability and actively assists by sharing traffic information vehicle control [23]. Although the information interaction between vehicles reduces the complexity and randomness of the environment, it does not essentially solve the complexity of rule-based design, nor can it continuously interact with the environment through data self-learning [24].

The rapid development of data-driven and artificial intelligence technology has received widespread attention in the field of transportation [1], and many researchers have provided machine learning methods to learn human driving habits. Wang et al. [13] proposed a DNN-based CF model using Next Generation Simulation (NGSIM) data. Wei et al. [14] added a supervised network trained by real driving data to an actor-critic network and proposed a car-following framework for supervised reinforcement learning. Wang et al. [15] proposed a human-like maneuver decision-making method based on an LSTM network and a conditional random field model for AVs. The above research shows that a data-driven model has high accuracy in fitting human driving trajectories, which significantly reduces the interference of developers in the strategy. However, it essentially only simulates human driving behavior rather than optimizing safety, efficiency, and comfort, and it is difficult to obtain an optimal driving strategy.

By comparison, DRL can adaptively update control strategy parameters by interacting with the environment. Deep Q-network (DQN) and its derivatives, a combination technique of Q-learning and large-scale nonlinear neural networks, have been presented in recent years to solve the vehicle decision control problem. Xia et al. [25] adopted a DQN algorithm to propose a driving strategy based on professional driver experience, which only relies on an image input of a camera to achieve end-to-end control. To solve the problem of DQN overestimation, Nageshrao et al. [26] proposed to use a dueling deep Q-network (DDQN) to learn driving strategies and safety checks to constrain actions. However, these approaches output discrete actions inefficiently in solving high-dimensional action space problems. More importantly, the action is continuous and precise in terms of AV control. In order to solve this continuous control problem, Lillicrap et al. [27] proposed a deterministic policy gradient (DPG)-based actor-critic, a model-free algorithm for continuous action space control. In an open racing car simulator (TORCS), Sallb et al. [28] used the same driving scenario to compare the driving strategies of DQN and DDPG. The results showed that a driving strategy based on DDPG completes a driving task more accurately and smoothly than a driving strategy based on DQN. In order to avoid making unpredictable decisions in the learning process based on historical driving data, Xiong et al. [29] designed a safety mechanism based on artificial potential fields by using DDPG to learn driving strategies. The research mentioned above makes progress in solving certain conditions of CF driving. However, the end-to-end decision-making strategy with images as input leads to insufficient driving status obtained by the DRL network. Moreover, the driving strategy learned in a single-environment training scenario is difficult to apply directly to the natural driving environment. Obtaining driving data through a vehicle test bench, Sun et al. [30] proposed a DDPG-based decision-making strategy of ACC for heavy vehicles. However, dividing a two-dimensional action space of acceleration and braking into a one-dimensional independent training action space results in an unacceptable driving situation in which acceleration and braking are output at the same time. Moreover, the fixed value punishment term given to the completed conditions, such as too many lane departures and collisions during training, is not conducive to the agent absorbing adverse experiences and accelerating the convergence of the network. For AVs, the comfort of passengers must be accepted in addition to the safety and efficiency of the vehicle. Zhu et al. [31] proposed a safe, efficient, and comfortable DRL-based speed control method. It obtains a fixed acceleration range through NGSIM data and designs a collision avoidance strategy in the face of an emergency, that is, braking at the maximum deceleration, but the occurrence of a collision is a fixed penalty. From the perspective of shaping the reward function, Pan et al. [32] collected and analyzed real-world car-following test data and developed a DDPG car-following model with a human-like reward function. Nevertheless, the reward function is all negative, there is no positive reward, and the punishment for collision is −1. Similarly, Yan et al. [33] combined the advantages of cooperative adaptive cruise control (CACC) and DDPG in car-following decision-making to output an optimal policy that also contains all negative reward functions, and there is no punishment term for training completed early.

The existing related work uses the DRL method to achieve vehicle decision-making control, and the DDPG algorithm solves the continuous problem in the field of vehicle control very well. In addition, most of the actions applied in the DRL work are fixed empirical constraints, and there are few considerations about how to use the driver’s acceleration characteristics in the car-following strategy. Therefore, we first analyze NDD to extract the driving characteristics of the car-following driver and determine the state space via correlation analysis. Then, according to the speed-acceleration distribution characteristics, the corresponding curve is fitted via 3σ boundary points of the normal distribution to enhance random exploration of the DRL action output. Finally, a multiobjective reward function combines safety, efficiency, and comfort. In order to create a CF model that can faithfully simulate a driver’s following behavior and that employs the DDPG method to solve the DRL problem, this work further studies the application of DRL to the modeling of the autonomous CF decision-making problem.

## 3. Analysis of Car-Following Behavior-Based Naturalist Driving Data

### 3.1. Source of Naturalist Driving Data

This paper utilizes the following driving characteristics from real-world microscopic driving data and combines them with DRL to realize autonomous following decision control. NGSIM data are widely used in the field of traffic flow [34], which plays a vital role in the analysis of driving behavior at the tactical level, especially in research into vehicle interaction behavior in acceleration and lane change models. Among them, the I-80 dataset collects 45-min vehicle trajectory data in three periods, representing the process from noncongestion to congestion and the peak period of traffic congestion, respectively. According to relevant research work using these data [13,31,35], we have established 1341 pairs of CF events, including FV speed and acceleration, speed of leading vehicle (LV), relative speed, and space headway. Moreover, two longitudinal safety parameters are added in the CF process, which are usually used for driver assistance systems, THW, and time to collision (TTC) [12]. In order to avoid the situation that space headway is zero and TTC tends to infinite in the above driver-following trajectory analysis, inverse time to collision (TTCi) is used instead of TTC analysis.

The CF events are used to analyze the driver’s naturalist driving behavior during the CF process, and the DRL agent is used for training and testing. Among them, 70% (939) of the CF events were selected as the training dataset, and the remaining 30% (402) as the test dataset by random sampling algorithm. At the same time, the high-occupancy lane in the US101 data is added for a single-scenario test and analysis.

### 3.2. Statistical Feature Analysis

In this section, we analyze the distribution of characteristic parameters based on the 1341 groups of CF events to know the driver’s behavior characteristics during the CF process. Figure 1 shows the empirical distribution of each characteristic parameter, including frequency and cumulative frequency.

For Figure 1a, the FV speed distribution in this dataset lies in the main scope of (6 m/s, 9.5 m/s), and 1% is greater than 18.6 m/s. This suggests that the FV mainly drives at medium and low speeds, and road traffic conditions are congested. From the distribution in Figure 1b, it can be seen that the acceleration of FV basically obeys a normal distribution and lies in the main scope of (−3 m/s^2^, 2.5 m/s^2^). This indicates that the driver maintains stable operation during the CF process, and there is basically no rapid acceleration or deceleration. In the acceleration process, 99.9% of values are less than 2.8 m/s, while in the deceleration process, 99.9% of values are less than 4 m/s. According to the actual testing data in the literature [8], the results showed that the maximum comfort deceleration value does not exceed 4 m/s^2^. Otherwise, it may cause discomfort to the occupant. From the frequency distribution and percentile of space headway in Figure 1c, it can be seen that around 1% is less than 40 m. If the driver keeps a short gap while following the car, it will help improve the utilization rate of the road. However, the excessive pursuit of a small gap can cause rear-end accidents easily, and it is also easy to cause psychological panic to the driver or passengers. As shown in Figure 1d, the distribution of CF relative speed basically conforms to a normal distribution, and the distribution range is [−2.5 m/s, 2.5 m/s]. The driver follows the LV with a slight speed difference. About 1% of the relative speed exceeds 2 m/s, and the maximum value is 2.5 m/s.

In the process of CF, the driver will make different decisions according to the motion relationship with the LV to maintain a safe driving state. THW and TTCi are used to evaluate whether decision behavior is safe. A smaller THW value indicates that the situation of the FV following the LV is more urgent, such as short space headway or high speed of a FV. From Figure 1e, the distribution of THW shows that the 70% distribution range is (1.3 s, 3 s), indicating that most drivers form a stable motion state with the LV, and about 1% is less than 0.5 s, which may be due to the higher speed of the FV or the minor space headway. TTCi is selected for the safety braking system, which is used to distinguish the driver in the dangerous state and the driver in the control state. The distribution in Figure 1f shows that the overall distribution of TTCi obeys a normal distribution, and only 0.6% is more significant than 0.25 s^−1^. According to the literature [12], a TTCi value of 0.25 s^−1^ is selected for safe collision avoidance.

### 3.3. Correlation Analysis

In order to further clarify which characteristic parameters are the main factors affecting the driver’s operating behavior, the decision-making basis for the driver’s decision-making behavior is established. Therefore, the Spearman correlation coefficient (Equation (1)) is used to analyze the correlation between the characteristic parameters [36].
(1)$$\rho =1-\frac{6\sum_{i=1}^{n}d_{i}}{n\left(n^{2}-1\right)}$$
where *d_i_* is the grade difference between the two variables, *n* is the number of samples, and the range of correlation coefficient *ρ* is (−1, 1). Among these, the positive and negative values indicate that the two variables are positively and negatively correlated. It is generally believed that the absolute value of ρ is less than 0.4, and the correlation between the two variables is weak. The two variables are highly correlated when the absolute value of ρ is greater than 0.7. The correlation coefficients between the characteristic parameters and the speed of FV are calculated, respectively, and the *p*-value of the significance tests is obtained. Figure 2 shows the frequency and significance probability distribution of the correlation coefficients among the parameters.

In Figure 2a, it can be seen that the probability that the absolute value of the correlation coefficient between each parameter and the FV speed is greater than 0.4 is more than 50%, and LV is the highest with that of FV, followed by space headway. In the process of CF, FV speed is positively correlated with the speed and space headway of LV, while FV speed is negatively correlated with space headway, THW, and TTCi, respectively. In order to better reflect the correlation between each characteristic parameter and the speed of FV, the distribution of the correlation coefficient is listed in Table 1. From the probability distribution of the correlation significance test in Figure 2b, it can be seen that except for TTCi (67%), the distribution probability of other parameters with a p-value of less than 0.05 exceeds 86%. It can be judged that LV speed, relative speed, space headway, and THW have a specific impact on the CF process.

## 4. Deep Reinforcement Learning for Autonomous Car-Following Decision-Making

### 4.1. State Space

The state space is the information FV uses to determine what will happen, including the environmental and FV state. Moreover, the state space should not only fully characterize the characteristics of FV at a certain moment but also be directly related to the convergence of DNN in the algorithm. From the analysis in Section 3, it can be seen that the driver’s speed in the following process is significantly affected by the speed of LV, space headway, relative speed, and THW. At the same time, THW is related to space headway and speed. Therefore, the reference information *s_t_* = {*v*_FV_, *d*_rel_, *v*_rel_} is selected to represent the driver’s action at *t* time by selecting the speed of FV, space headway, and relative speed. In the process of autonomous decision-making following control, the agent refers to the decision-making algorithm to interact with the environment. According to the longitudinal kinematics characteristics between FV and LV, the iterative relationship of the environmental state is described by a kinematic point mass model (Equation (2)):(2)$$\left\{ \begin{array}{l} v_{\mathrm{FV}}(t+1)=v_{\mathrm{FV}}(t)+a_{\mathrm{FV}}(t)\times T_{s}\\ v_{\mathrm{rel}}(t+1)=v_{\mathrm{LV}}(t+1)-v_{\mathrm{FV}}(t+1)\\ d_{\mathrm{rel}}(t+1)=d_{\mathrm{rel}}(t)+\frac{v_{\mathrm{rel}}(t)+v_{\mathrm{rel}}(t+1)}{2}\times T_{s} \end{array}\right.$$
where *d*_rel_ is the space headway, *T_s_* is the sampling period, *v*_FV_ is the speed of the following vehicle, *a*_FV_ is the acceleration of the following vehicle, *v*_rel_ is the relative speed, *v*_LV_ is the speed of the leading vehicle, and *v*_rel_ is the difference in speed between the following vehicle and the leading vehicle. The current moment and the next moment is represented by *t* and *t +* 1, respectively.

### 4.2. Action Space

In most applications of DRL, the agent’s actions are constrained by fixed experience without considering the driver’s dynamic characteristics. Therefore, to realize autonomous following decision-making and enhance the exploration ability of the decision-making algorithm, the dynamic relationship between the speed and acceleration of FV is established with 939 pairs of CF events in the training dataset, as shown in Figure 3. It is evident from the figure that the distribution of the data points is dense, sparse, and between two sides, and the acceleration/deceleration value decreases with an increase in speed. In the low-speed driving range (*v*_FV_ ≤ 11 m/s), the acceleration distribution is very dense, accounting for 87.5% of the training dataset. In the medium-speed range (11 m/s ≤ *v*_FV_ ≤ 21 m/s), the acceleration distribution is relatively dense, accounting for 12.3% of the training dataset. The acceleration distribution is sparse in the low-speed driving range (*v*_FV_ > 21 m/s), accounting for 0.2% of the training dataset. Consequently, the normal distribution of the 3σ boundary points of each data part is counted according to the density interval. The curve fitting toolkit obtains the dynamic response curves of velocity and acceleration.

The decision of action should be changed from a deterministic process to a random process. Then, the action is sampled from this random process and passed to the environment interaction. In order to make the FV have more CF decision-making, Gaussian noise is added to the policy output of the policy network to make it randomly sample and explore in the speed-acceleration distribution area. Therefore, the actual output action is *a_t_* = ℕ(*a*, σ^2^), as shown in Figure 4.

Here, σ^2^ is the variance in Gaussian noise and is reduced by the decay rate *ξ* in each training step, which can be expressed as Equation (3):(3)$$\sigma_{t+1}=\xi \sigma_{t}$$
where *ζ* is the decay rate in each training step and σ is the variance in Gaussian noise.

### 4.3. Reward Function

The reward function guides the adjustment direction of the parameters of DNN so that the output action can make FV perform as desired. Hence, the design of the reward function affects the decision-making performance of FV. In a real traffic environment, a vehicle controlled by a driver will take acceleration or deceleration action to adjust the vehicle’s longitudinal motion state based on the driving environment so that the speed and gap of the vehicle are within an acceptable, safe, and comfortable zone. In order to better reflect the characteristics of the driver’s CF behavior, autonomous CF decision-making is achieved. The multiobjective reward function is designed by referring to the driving task, such as safety, efficiency, and comfort. Thus, the principle of reward function is as follows:

**(i) Safety:** As the most basic and essential control purpose, it directly affects vehicle and passenger life and property safety. According to the relevant data [37], rear-end collision is the most frequent traffic accident in driving. TTC indicates vehicle crash risk, and smaller TTC values correspond to higher crash risk and vice versa. Therefore, to avoid the case of small TTC to improve driving safety, a too-small TTC value is given great punishment, where the logarithmic function conforms to this feature. For the CF driving task, this paper chooses TTCi instead of TTC as the safety evaluation parameter, so the constructed safety reward function is as Equation (4)
(4)$$r_{s}(t)=\left\{ \begin{array}{l} \log(\frac{TTCi^{*}+\alpha }{TTCi(t)})\;\;\;TTC(t)\geq TTC^{*}\\ 0\;\;\;\;\;\;\;\;\;\;\;\;\;\;\;\;\;\;\;\;\;\;\;\;\;\;\mathrm{otherwise} \end{array}\right.$$
where *TTCi* is the inverse time to collision, *TTCi** represents the threshold of *TTCi*, α is the weight parameter, and *r_s_* is the constructed safety reward.

**(ii) Efficiency:** Under the guarantee of the safe driving of AVs, improving road utilization is directly reflected in achieving the desired gap by adjusting its speed. Thus, THW is used to represent the driving efficiency of the vehicle. According to the THW probability distribution, an appropriate reward mapping relationship is determined. Figure 5 shows the THW probability density distribution in the training dataset. The data are fitted by the normal distribution function, lognormal distribution function, and kernel density estimation (KDE) function [38]. Obviously, the fitting effect of KDE is closer to the actual THW distribution. Especially at the maximum probability density of THW, the probability density value of KDE is 4.6% more than that of the lognormal distribution, the corresponding THW is about 7%, and the fitting effect of the normal distribution is the worst. For the CF driving task, the constructed efficiency reward function is expressed as Equations (5) and (6):(5)$$r_{e}(t)=f_{KDE}\left[THW(t)\right]$$
(6)$$f_{KDE}=\frac{1}{nh}\sum_{i=1}^{n}K(\frac{x-x_{i}}{h})$$
where *K* (·) is the kernel function, *n* is the amount of data observed, *h* is the bandwidth, and *x_i_* is the sample point of the independent distribution. The Gaussian function is selected as the kernel function of KDE.

**(iii) Comfort:** Jerk is an essential indicator for evaluating ride comfort, which is determined by acceleration variation [39]. By constraining the jerk change in the driving process, the great inertia impact from driving brought by a vehicle to its passengers will be reduced, improving ride comfort and reducing fuel consumption. For the CF driving task, the constructed passenger comfort reward function is expressed as Equation (7):(7)$$r_{c}(t)=\beta {\left[a_{\mathrm{FV}}(t)-a_{\mathrm{FV}}(t-1)\right]}^{2}$$
where *β* is the weight parameter, *a*_FV_ is the acceleration of the following vehicle, and *r_c_* is the constructed passenger comfort reward. Considering the above three driving levels to build the reward function, the agent is not very intelligent in trial-and-error learning and does not learn from the error events. For example, in the training process, only the training process is stored for collision, or extremely conservative collision avoidance leads to stopping. Therefore, in order to make the agent learn adverse experiences and accelerate the convergence of the network, a kinetic energy penalty reward (such as collision) and a potential energy penalty reward (such as early stop) are introduced, respectively. In the training process, the penalty reward of kinetic energy in the form of collision is expressed as Equation (8):(8)$$r_{K}\left(t\right)=\delta \left[v_{\mathrm{FV}}{\left(t\right)}^{2}\right]1\left(\mathrm{done}=\mathrm{collision}\right)$$
where *δ* is the weight parameter, *v*_FV_ is the speed of the following vehicle, and *r*_K_ is the penalty reward of kinetic energy. The term **1** (done = collision) means that the value is 1 when FV collision occurs; otherwise, it is 0. The penalty reward in the form of potential energy for the early stop is as Equation (9):(9)$$r_{P}\left(t\right)=\varepsilon \left[d_{\mathrm{rel}}{\left(t\right)}^{2}\right]1\left(\mathrm{done}=\mathrm{over}\right)$$
where *ε* is the weight parameter, *d_rel_* is the space headway, and *r*_P_ is the penalty reward in the form of potential energy. The term **1** (done = over) means that the value is 1 when FV early stop occurs; otherwise, it is 0. In summary, the overall reward function is the above linear combination as per Equation (10):(10)$$r(t)=Normal[r_{s}(t)+r_{e}(t)+r_{c}(t)+r_{K}(t)+r_{P}(t)]$$

### 4.4. Termination Conditions

In order to avoid learning the optimal local strategy, if at least one of the following events occurs during the training process, the episode ends and enters the next episode of the reset environment state.

(i)Collision: the FV is not effectively braked, resulting in traffic accidents.(ii)Early stop: the FV has too conservative collision avoidance, leading to stopping.(iii)Vehicle stuck: the FV speed is always lower than 0.1 m/s within 10 steps.(iv)No reward increase: no increase within 100 steps in each episode.

### 4.5. CF Decision-Making Algorithm

This study combines the DDPG algorithm with the driver’s behavior characteristics to learn the optimal driving strategy. Figure 6 shows an agent-environment interaction model for autonomous car-following decision-making control, i.e., the interaction between the following tasks, driving characteristics, and traffic information. Through extensive real-world, data-driven decision model training, the actor network receives state *s_t_* and outputs deterministic policy. After obeying the Gaussian distribution and training dataset speed-acceleration constraint boundary, the actor network outputs action to achieve CF’s purpose. The actor network parameter update follows the deterministic policy gradient theorem as per Equation (11):(11)∇θμJ(μ)≈1N∑t[∇aq(st,a)|a=μ(st)+ℕ⋅∇θμμ(st)]
where *N* is the time range of the sampling time, *θ^μ^* denotes the policy parameters, *μ(s_t_)* is the deterministic policy, q(st,a)|a=μ(st)+N is the action-value function, *a* is the actor’s action in the actor network, and *s* is the current state.

The update method of the critic network is to minimize loss, as per Equation (12), and the term *y_t_* is derived from the critic target network and actor target network (Equation (13)).
(12)$$L(q)=\frac{1}{N}\sum_{t}\left[y_{t}-q(a_{t}\right){]}^{2}$$
(13)$$y_{t}=r_{t}+\gamma q^{\prime }({a^{\prime }}_{t+1})$$
where *y_t_* is the current real reward value, *r_t_* is the overall reward of the present moment,  γ*′* is the discount factor of the future reward value, *q′(a′_t+1_)* is the action-state value function corresponding to the next moment, *L(q)* is the loss function, and *q(a_t_)* is the action-value function of the current moment.

During each iteration, the actor-critic target network parameters are slowly approximated to the current actor-critic network parameters by the soft update (Equation (14)):(14)$$\left\{ \begin{array}{l} \theta_{\mu^{\prime }}\leftarrow \tau \theta_{\mu }+(1-\tau )\theta_{\mu^{\prime }}\\ \theta_{q^{\prime }}\leftarrow \tau \theta_{q}+(1-\tau )\theta_{q^{\prime }} \end{array}\right.$$
where τ is the update rate with τ << 1, and $\theta_{\mu }$ and $\theta_{q}\;$ are the parameters of the current actor-critic network. In this way, the network parameters change slowly, improving the learning process’s stability.

Since the input of the decision system does not need to be presented in the form of images, it only needs to obtain measurement information, such as the speed and space headway of the car-following event, which is combined in a vector to form an MDP state *s_t_* at time *t*. Thus, we decide to use DNN instead of the traditional CNN structure, which can significantly simplify the network and reduce computational burden. Additionally, the architecture of the actor-critic network shown in Figure 7 is designed according to the car-following tasks. Considering the problem’s complexity, convergence rate, and computational complexity, we use a multilayer DNN, and the network size decreases layer by layer. In an actor network, the hidden layer is 3, and the number of nodes in each hidden layer is 64, 48, and 24, respectively. The activation function is ReLU, and the output layer is Tanh. In the critic network, the hidden layer structure is the same as the actor, but the output layer is linear activation.

## 5. Simulation Results and Discussion

### 5.1. Simulation Setup

To verify the effectiveness and accuracy of our proposed car-following strategy, a simple numerical car-following model is implemented. The developed strategy is evaluated through extensive simulations. Each episode of the training process is randomly selected from the training dataset, and the selected first row of the car-following event is taken as the initial state, i.e., *v*_FV_ = data_n_(0,0), *d*_rel_ = data_n_(0,1), and *v*_rel_ = data_n_(0,2). The FV learns the deterministic policy from trial and error to achieve continuous control and then iterates to generate FV speed, relative speed, and space headway at the next moment based on Equation (4). Table 2 shows the detailed parameters of the training.

### 5.2. Simulation Results

**(i) *DRL learning efficiency:*** We first evaluate the learning ability of the DRL method. Inspired by the control variable method, we designed three similar DRL strategies (i.e., reference action space and reward functions) and an MPC-based approach to compare our car-following decision-making strategy. The following are the simulation results and discussions on different aspects. Under the same simulation conditions, we use the following strategies to compare simulation performance:(a)We use the DDPG algorithm combined with NDD to achieve an autonomous car-following decision-making strategy. The penalty reward in the form of mechanical energy is introduced in the design of the reward function, which is a function of speed and space headway rather than constant reward. Meanwhile, the 3σ boundary of the speed-acceleration fitting curve of the training dataset is used to realize varying constraints for FV action (recorded as our proposal).(b)In the application of some DRL algorithms, the action output by the agent generally uses fixed empirical constraints. Thus, the fixed empirical constraint (FEC) action range of FV is determined by referring to NDD, i.e., [*a*_min_, *a*_max_] = [−4 m/s^2^, 2 m/s^2^], and the other parameters are the same as our proposal (recorded as FEC).(c)In some DRL studies, constants are used as punishment rewards for collision or lane departure in agent training. Hence, a constant value for FV collision and early stop is used as the reward function, i.e., Equation (8) is changed to *r*_k_(t) = −100, and Equation (9) is changed to *r*_p_(t) = −50. Furthermore, FV’s action also uses fixed empirical constraints (recorded as FEC w/CP).(d)The DRL strategy established uses the same varying constraint as our proposal and constants as the reward function of collision and early stop (recorded as VC w/CP).(e)A rule-based control strategy is established regarding the characteristics of car-following behavior, combined with safety, efficiency, and comfort as multiobjective constraints. An MPC car-following model based on constant THW is constructed, in which the model parameters are determined according to the distribution of the characteristic parameters of car-following behavior. Similarly, the reward function is designed similarly to our proposal (recorded as MPC-based).

The efficiency performance evaluation results are shown in Figure 8, which shows the reward obtained by FV and training events under different strategies. It can be seen from Figure 8a that with an increase in the episode, the established DRL-based learning strategies gain a gradually stable return (i.e., from negative to positive values). This suggests that autonomous driving strategies have been well learned, maximizing long-term rewards. However, the rewards of the FEC and VC w/CP strategies fluctuate significantly in each episode, and the learned driving strategies are unstable. The real-world car-following data established in this paper are complicated and dynamic, enabling autonomous decision-making. The reward function contains a nonconstant penalty reward, and the reward value of each episode is larger (value less than −100) before training. This is because FV has no driving experience during the initial training period, and the probability of action exploration is larger, resulting in more rear-end collisions and early stop events. Our proposal contained 395 collisions and 141 early stops, and 47 car-following events were completed in the collision training episode. There were 315 collisions, 174 early stops, and 58 car-following events in the collision training episode for FEC. To better reflect learning during strategy training, Table 3 lists the overall results. By comparing the reward values during training, we find that our proposal is relatively the best. From Figure 8b, the mean reward value of our proposal is greater than 0.5 (in episodes 683 to 729). In the next 300 episodes, the agent explores and exploits actions to learn the optimal driving strategy, resulting in a fluctuation of the reward value in this part of the training episode. After the 1062nd episode, the mean reward value is greater than 0.5, and the average reward value in the remaining training episode is 0.61. For FEC, the mean reward value at the beginning of training is similar to our proposal, and its mean reward value at the 647th episode is 0.57. However, with the training process value decreasing, the 883rd episode began to be less than 0.5 until it trended to 0.01. For constant penalty terms, the episodes of mean reward greater than zero are shorter, but VC w/CP is more volatile, the values are small, and the mean reward value fluctuates around 0.18. For FEC w/CP, the mean reward is greater than 0.5 at the 410th episode, but the extreme deviation in fluctuation in the remaining episodes is 0.13. The mean reward for MPC-based fluctuates around 0.52. However, for fixed empirical action and constant strategy, FV can obtain a larger reward value in the middle of the training, but the stability of the strategy is poor, with an increase in the episodes that cannot be well explored and used for DRL action. Since the reward function guides the adjustment direction of DNN parameters, the output action enables FV to perform as desired. The constant penalty makes similar decisions for different termination events, resulting in poor stability in learning driving strategies.

Next, to verify the strategy’s effectiveness, we take the car-following events of the test dataset as the vehicle trajectory input and compare the reward values of the car-following events obtained by the saved policy models. All strategies did not collide during the whole test. Figure 9 shows the reward values of each car-following event in the test dataset. As seen from the figure, our proposal and FEC w/CP achieve the smallest reward fluctuation, and our proposed value is more concentrated (only one peak), while MPC-based and VC w/CP obtain the largest fluctuation. For the average reward value of all test car-following events, our proposal is 313% higher than FEC, 32% higher than FEC w/CP, 226% higher than VC w/CP, and 19% higher than MPC-based, respectively.

**(ii) *Decision-making performance:*** To illustrate our approach to autonomous decision-making, FV safely, efficiently, and comfortably follows LV. Firstly, we choose our proposal, FEC w/CP, and MPC-based decision-making models to analyze the decision-making performance of different strategies from a car-following event. Then, our proposal is used to simulate all car-following events in the test dataset and analyze the characteristic parameters generated by the FV decision. Finally, we compare our statistics with the distribution of the test dataset.

***Single scenario:*** The training process’s minimum reward value (−464.66) is used for vehicle decision-making performance analysis as the car-following event. The initial space headway is 19.27 m, the LV speed is 7.5 m/s, and the FV speed is 9.7 m/s. Figure 10 shows the decision performance of different strategies in the single-scenario training dataset. It can be seen from the figure that the changing trends in FEC and our proposal in space headway, speed, and THW curve basically coincide, but FEC acceleration has an obvious continuous step change at 1.2 s until the end of the car-following event, resulting in a large inertial impact (i.e., step change in jerk) in the car-following process. The VC w/CP method adopts a more conservative car-following strategy because, at the beginning of the process, the vehicle decelerates at −4 m/s^2^, the acceleration curve change trend is basically consistent with our proposal after 5 s, and all THW values are greater than 2 s. At the end of the car-following event, the shortest space headway between our proposal is 7.88 m, which is 30% smaller than human, 36% smaller than MPC-based, 2% smaller than FEC, 22% smaller than FEC w/CP, and 520% smaller than VC w/CP. Additionally, after about 5 s, our proposal THW values basically stabilize at about 1.2 s for car following, while FEC w/CP basically stabilize at around 1.43 s after roughly 7 s. At the beginning of the acceleration curve change, our proposal, FEC, and FEC w/CP strategies all accelerate (*a*_initial_ = 2 m/s^2^) to approach LV, while the human and MPC-based strategies decelerate (*a*_initial_ = −4 m/s^2^) and VC w/CP also decelerates (*a*_initial_ = −1.5 m/s^2^). For safety analysis, the TTCi value of our proposal exceeds the threshold in the previous 2 s, and the maximum value exceeds the threshold by 11% (*TTCi*_max_ = 0.28 s^−1^). For FEC, the value of TTCi only exceeds the threshold in the previous 2.5 s, and the maximum value exceeds the threshold by 31% (*TTCi*_max_ = 0.36 s^−1^).

For a single scenario in the test dataset, the initial state of the space headway is 70.45 m, the FV speed is 21.03 m/s, and the LV speed is 15.57 m/s. The single-scenario decision-making performance of the five decision-making models is shown in Figure 11. As can be seen from the figure, the changing trend of each performance index curve is basically consistent with the single scenario in the training dataset. Among them, the changing trend of FEC and FEC w/CP is similar to that of our proposal curve, but FEC acceleration has an obvious continuous step change in 8 s until the end of the car-following event, resulting in a large inertial impact (i.e., step change in jerk) in the car-following process. At the end of the car-following event, the shortest space headway between our proposal is 22.81 m, which is 67% smaller than human, 45% smaller than MPC-based, 9% smaller than FEC, 32% smaller than FEC w/CP, and 50% smaller than VC w/CP. Moreover, after about 7 s, our proposal THW values basically stabilize at about 1.16 s for car following, while FEC w/CP basically stabilize at about 1.7 s. At the beginning of the acceleration curve change, our proposal, FEC, FEC w/CP, and MPC-based strategies all accelerate (*a*_initial_ = 2 m/s^2^) to approach LV. In contrast, the human and VC w/CP strategies decelerate (*a*_initial_ = −4 m/s^2^). For safety analysis, the TTCi value of our proposal is the same as the threshold in the previous 4.5–5 s. For FEC, the value of TTCi only exceeds the threshold in the previous 4.3–6 s, and the maximum value exceeds the threshold by 19% (*TTCi*_max_ = 0.31 s^−1^).

For a single scenario in US101 (high-occupancy lane), the initial state of the space headway is 50.4 m, the FV speed is 15.23 m/s, and the LV speed is 14.45 m/s. The single-scenario decision-making performance of the five decision-making models is shown in Figure 12. As can be seen from the figure, the changing trend of each performance index curve is basically consistent with the single scenario in the training and test dataset. Among them, the changing trend of FEC and FEC w/CP is similar to that of our proposal curve, but FEC acceleration has an obvious continuous step change in 10 s until the end of the car-following event, resulting in a large inertial impact (i.e., step change in jerk) in the car-following process. At the end of the car-following event, the shortest space headway between our proposal is 20.11 m, which is 38% smaller than human, 28% smaller than MPC-based, 8% smaller than FEC, 32% smaller than FEC w/CP, and 54% smaller than VC w/CP. Moreover, after about 8 s, our proposal THW values basically stabilize at about 1.17 s for car following, while FEC w/CP basically stabilize at about 1.7 s after about 11 s. At the beginning of the acceleration curve change, our proposal, FEC, FEC w/CP, and MPC-based strategies all accelerate (*a*_initial_ = 2 m/s^2^) to approach LV. In contrast, the human and VC w/CP strategies decelerate (*a*_initial_ = −0.86 m/s^2^, −0.14m/s^2^). For safety analysis, only the value of FEC at 6.3 s–6.4 s TTCi is the same as the threshold value.

In conclusion, based on the test analysis, our proposal can learn a safer, more efficient, and more comfortable car-following strategy. Among them, the changing trend of FEC and FEC w/CP is similar to our proposal, but FEC produces continuous step acceleration when the space headway is small, resulting in a large inertial impact. VC w/CP is a very conservative car-following strategy, with a space headway of about 40 m from the LV. The decision-making performance of MPC-based in the test scenarios with small space headway and low-speed driving is close to our proposal, but significant space headway and high-speed driving are poor.

***All scenarios:*** In this section, we design a multiobjective reward function that is secure, effective, and comfortable to compare human driving performance with that of our proposed car following.

**(1) Safety:** Instead of TTC, we utilize TTCi to assess driving safety. From the TTCi distribution in Figure 13, it can be seen that the TTCi distribution of the five strategies presents the characteristics of normal distribution, and the distribution is more concentrated than that for humans. In all test scenarios, the percentiles corresponding to TTCi* for human, MPC-based, FEC, FEC w/CP, VC w/CP, and our proposal are 27.1%, 99.98%, 99.9%, 99.98%, and 99.3%, respectively. Therefore, it is possible to achieve FV driving safely by using DRL to create an autonomous car-following decision-making strategy.

**(2) Efficiency:** We choose THW to represent the driving efficiency of AVs and use KDE to fit the THW probability density distribution curve in the training dataset to design the reward function. From the THW distribution in Figure 14, it can be seen that the distribution range of FEC, FEC w/CP, and our proposal is smaller and concentrated and can make THW tend to a fixed value during the car-following process. In all test scenarios, THW values less than 1.2 s account for 27.1%, 19.8%, 22.6%, 0.7%, 4.4%, and 43% of human, MPC-based, FEC, FEC w/CP, VC w/CP, and our proposal, respectively. The overall results for different efficiency levels are shown in Table 4.

**(3) Comfort:** We enhance ride comfort by constraining the jerk change during driving. From the jerk distribution in Figure 15, it can be seen from the jerk distribution in Figure 9 that the distributions of MPC-based, FEC w/CP, VC w/CP, and our proposal show obvious normal distribution characteristics. The distribution range of FEC w/CP, VC w/CP, and our proposal is smaller and concentrated. The distribution range of FEC is mainly around [−60 m/s^3^, −40 m/s^3^], 0 m/s^3^, and [40 m/s^3^, 60 m/s^3^]. In all test scenarios, the proportion of an absolute impact value of less than 1.5 m/s^3^ is 56.5%, 85%, 59.1%, 97.9%, 98.6%, and 92% for human, MPC-based, FEC, FEC w/CP, VC w/CP, and our proposal respectively. The overall results of different comfort-level ratios are listed in Table 5.

## 6. Conclusions

This paper proposes an agent-environment interaction model of an autonomous car-following decision-making model to provide automatic driving that is safe, effective, and comfortable. Firstly, the distribution of speed-acceleration is established according to NGSIM data, and the corresponding curve is fitted according to the 3σ boundary point of a normal distribution to realize the variable constraint of the agent’s actions. However, most research into applied deep reinforcement learning uses fixed constraints on actions. Secondly, a safe, efficient, and comfortable multiobjective reward function for the automatic driving task is designed. A punishment term in kinetic and potential energy is introduced to make the agent remember the adverse experience, making the training agents perform better. The extensive simulation results show that our proposal can learn autonomous driving strategies through real-world driving data, which is significantly better than human driving. For future work, we will collect enough individual drivers’ driving behaviors as historical data to further train the model to serve personalized driving.

## Figures and Tables

**Figure 1 sensors-22-08055-f001:**
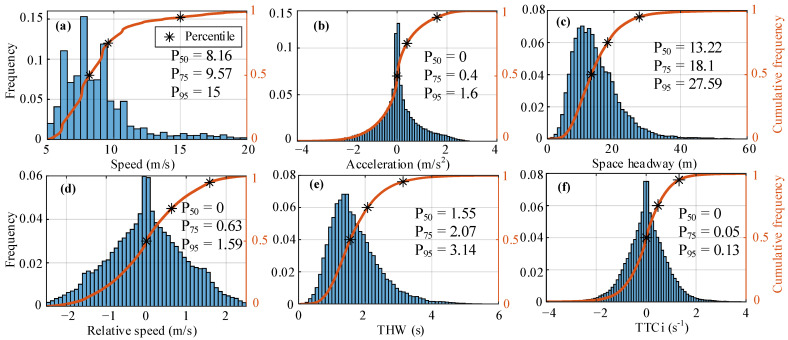
Empirical distribution of characteristic parameters in car-following events, where (**a**–**f**) respectively correspond to the following vehicle speed and acceleration, space headway, relative speed, time headway, and inverse time to collision.

**Figure 2 sensors-22-08055-f002:**
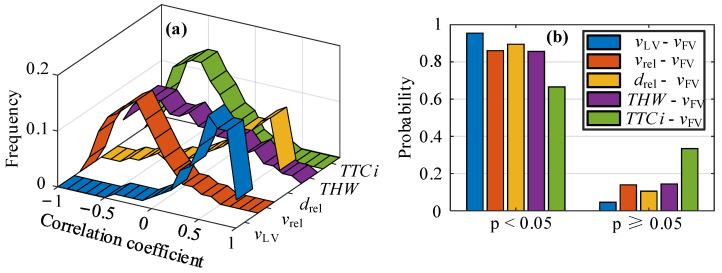
Distribution of correlation between parameters and following vehicle speed, where (**a**) shows the probability distribution of the maximum correlation coefficient, and (**b**) shows the *p*-values of the significance test.

**Figure 3 sensors-22-08055-f003:**
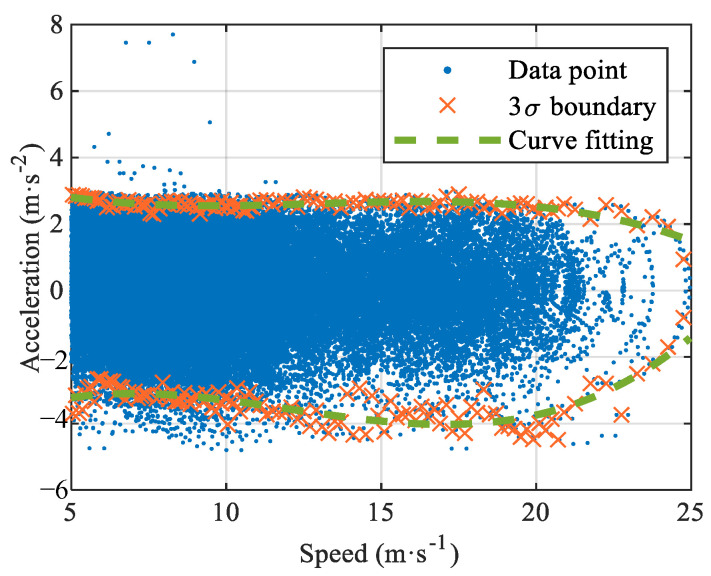
Dynamic characteristics of speed-acceleration based on training dataset.

**Figure 4 sensors-22-08055-f004:**
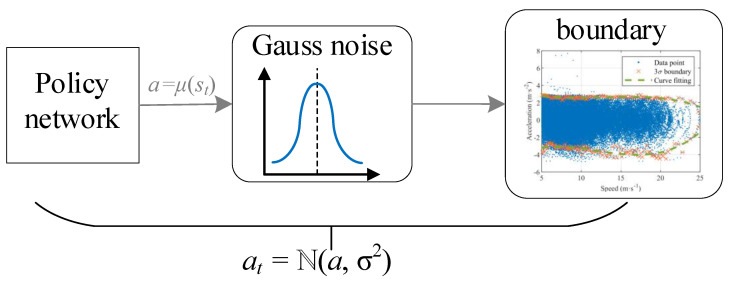
Random exploration and 3σ varying constraints for agent actions.

**Figure 5 sensors-22-08055-f005:**
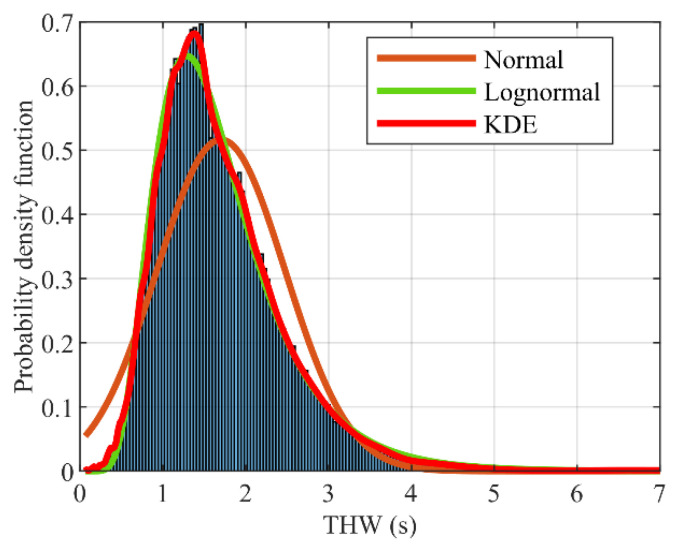
Time headway probability density distribution and fitting curve based on training dataset.

**Figure 6 sensors-22-08055-f006:**
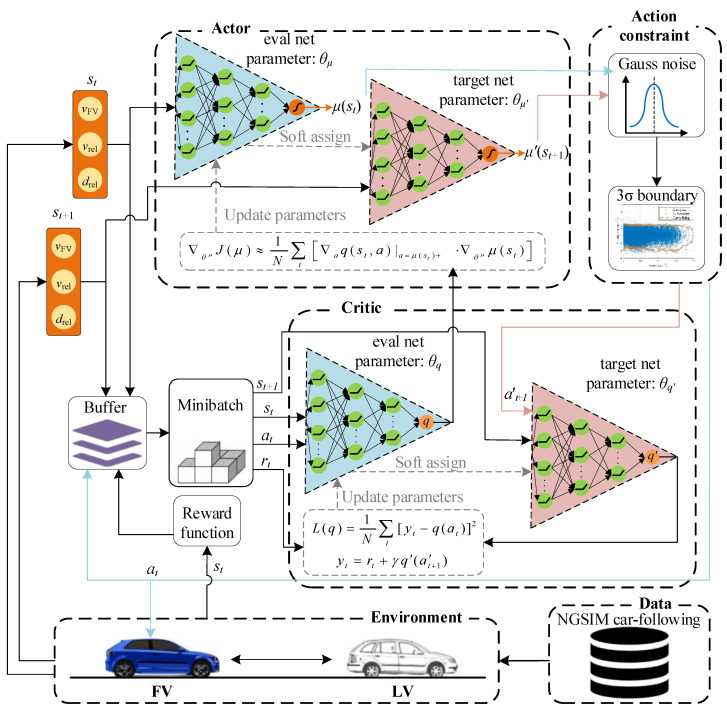
Agent-environment interaction model for autonomous car-following decision-making-based NGSIM data.

**Figure 7 sensors-22-08055-f007:**
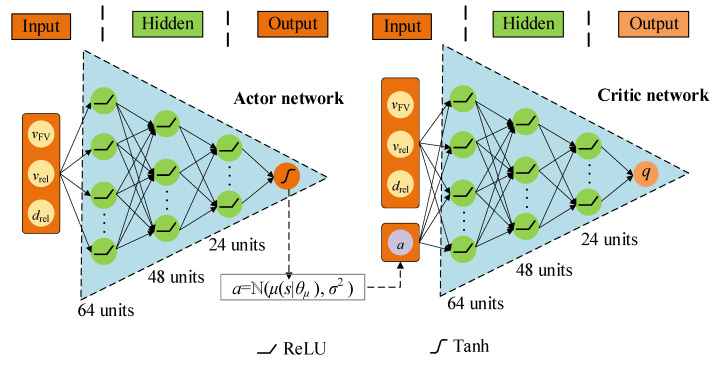
Structure of actor-critic network for solving autonomous car-following decision-making problem.

**Figure 8 sensors-22-08055-f008:**
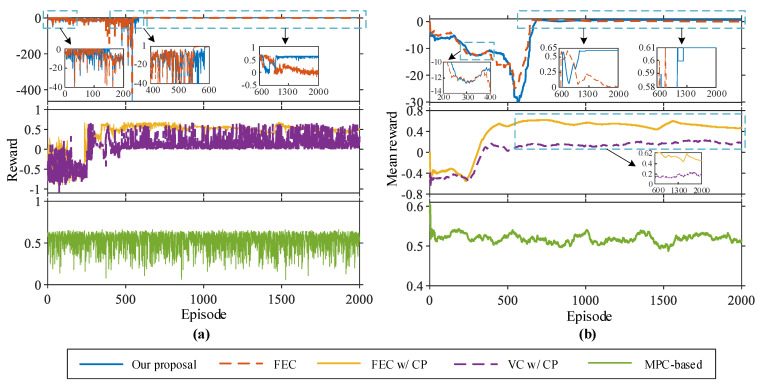
Changing of episode reward achieved during training, where (**a**) is episode reward for the five strategies, respectively, (**b**) is episode mean reward for the five strategies, respectively, and mean reward is the average of mean episode rewards across a rolling window with size 100.

**Figure 9 sensors-22-08055-f009:**
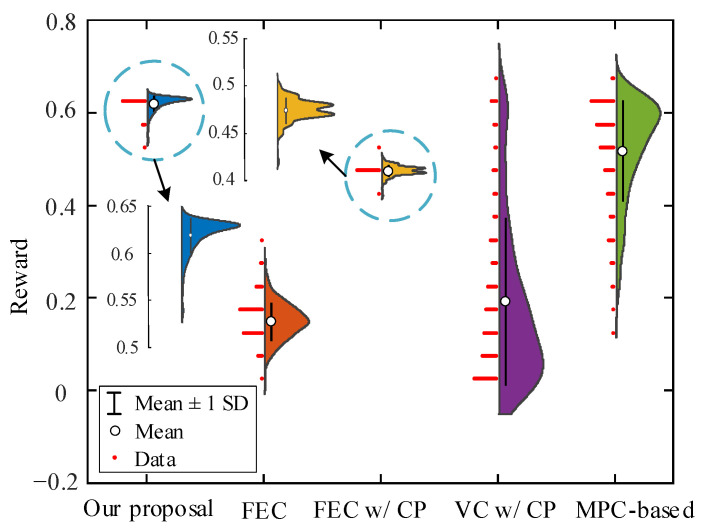
Distribution of episode reward achieved based on test dataset for different strategies.

**Figure 10 sensors-22-08055-f010:**
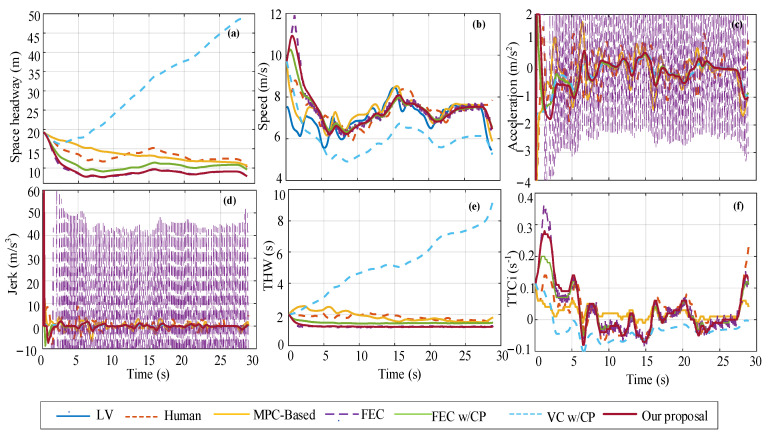
Comparison of decision performance of different strategies in single-scenario training dataset, where (**a**–**f**) correspond to space headway, speed, acceleration, jerk, THW, and TTCi, respectively.

**Figure 11 sensors-22-08055-f011:**
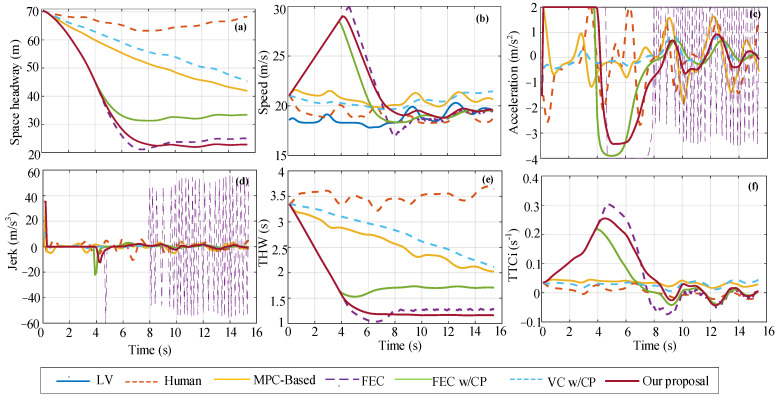
Comparison of decision performance of different strategies in single-scenario test set, where (**a**–**f**) respectively correspond to space headway, speed, acceleration, jerk, THW, and TTCi.

**Figure 12 sensors-22-08055-f012:**
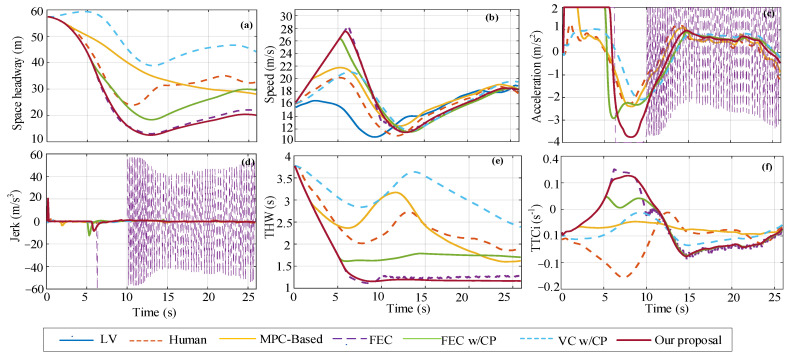
Comparison of decision performance of different strategies in single-scenario US101 set, where (**a**–**f**) respectively correspond to space headway, speed, acceleration, jerk, THW, and TTCi.

**Figure 13 sensors-22-08055-f013:**
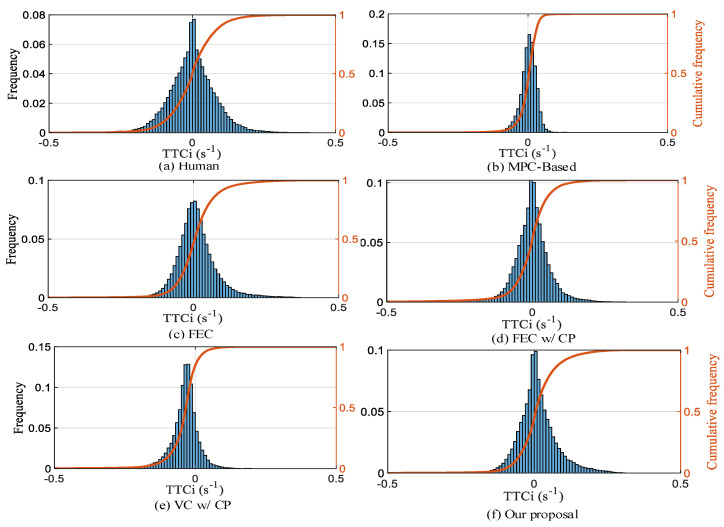
Comparison of TTCi frequency distribution for different strategies.

**Figure 14 sensors-22-08055-f014:**
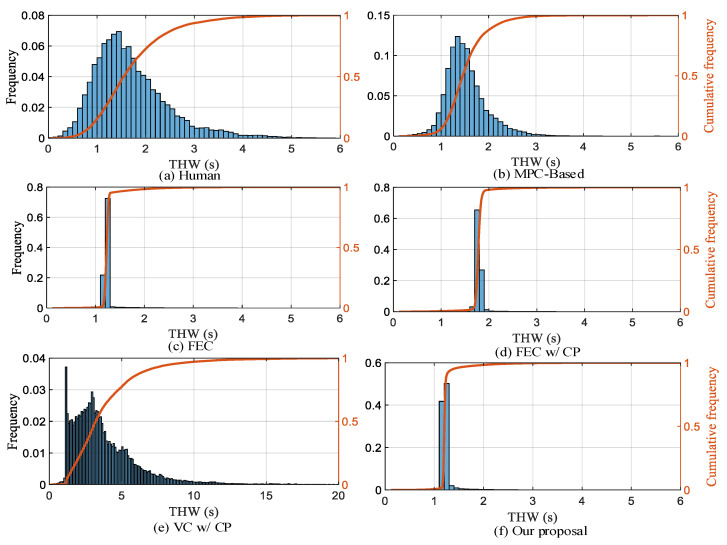
Comparison of THW frequency distribution for different strategies.

**Figure 15 sensors-22-08055-f015:**
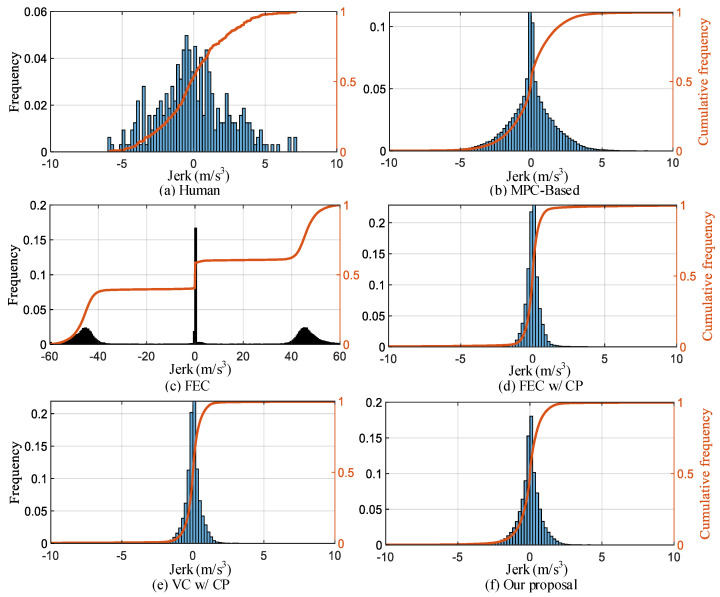
Comparison of jerk frequency distribution for different strategies.

**Table 1 sensors-22-08055-t001:** Ratio of correlation coefficient between each parameter and following vehicle speed.

Correlation	|*ρ*| > 0.4	|*ρ*| > 0.7
*v*_LV_-*v*_FV_	80%	39%
*v*_rel_-*v*_FV_	49%	11%
*d*_rel_-*v*_FV_	67%	32%
*THW*-*v*_FV_	55%	22%
*TTCi*-*v*_FV_	50%	11%

**Table 2 sensors-22-08055-t002:** Simulation parameter setting for autonomous car-following decision-making strategy.

Parameter	Value
Learning of actor network	0.0001
Learning of critic network	0.00001
Discounting factor of reward	0.9
Soft assign rate	0.001
Capacity of replay buffer	20,000
Size of minibatch	256
Decay rate	0.9995
Initial variance in the exploration space	3
Weight parameters: α, β, δ, ε	1 × 10^−5^, 0.028, 10, 5

**Table 3 sensors-22-08055-t003:** CF situations with different strategies in collision episodes achieved during training.

Strategy	Collision	Early Stop	Completion of CF Event During Collision
Our proposal	395	141	47
FEC	315	174	58
FEC w/CP	178	57	14
VC w/CP	205	46	29

**Table 4 sensors-22-08055-t004:** Ratio of efficiency for different strategies.

Strategy	*THW* ≤ 1.2	*THW* ≤ 1.5	*THW* ≤ 2
Human	27.1%	47.1%	72.6%
MPC-based	19.8%	54.7%	87.7%
FEC	22.6%	96.4%	98.4%
FEC w/CP	0.7%	1.1%	98%
VC w/CP	4.4%	10.6%	20.7%
Our proposal	43%	96.4%	98.4%

**Table 5 sensors-22-08055-t005:** Ratio of comfort for different strategies.

Strategy	|*Jerk*| ≤ 1.5	|*Jerk*| ≤ 2	|*Jerk*| ≤ 5
Human	56.5%	65.8%	94.5%
MPC-based	85%	90%	99.1%
FEC	59.1%	59.4%	60.1%
FEC w/CP	97.9%	98.3%	99.1%
VC w/CP	98.6%	99%	99.6%
Our proposal	92%	96.2%	99.2%

## Data Availability

Not applicable.

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
