# Peer review of "A Decision-Making Strategy for Car Following Based on Naturalist Driving Data via Deep Reinforcement Learning"

_sensors, 2022, doi:10.3390/s22208055_

Round 1

Reviewer 1 Report

The authors propose a car-following decision-making strategy using deep reinforcement learning using naturalist driving data. They analyze the CF driving behavior, speed-acceleration and conducted several simulations. 

Overall, the research is sound and the results were also well shown. Some minor edits in English and some figures are very hard to see. Some examples below.

Using twice? Line 622 On the one hand, and Line 624 On the other hand,

Would want to enlarge font sizes in figures (x, y axis and legend). especially in Fig 10, 11, 12.

Reviewer 2 Report

Review comments.

The article is well prepared and addresses a topical subject.  There are a number of suggestions to improve quality of the article.

1.       You need to spell out all abbreviations (e,g. CF) in the introduction at the first appearance of them, even if you have done so in the abstract.

2.       Please try to keep to the standard abbreviations. Car following models are not commonly referred to as CF?  if it is essential to include these abbreviations, then introduce them as discussed above.

3.       I suggest to add a table of all data sources and models with a column that include a very brief explanation of each.  For example, NGSIM data characteristics, CF, etc.   This will improve clarity of the paper and the approach.

4.       First sentence in introduction is a bit misleading; CF has not become the most common driving behavior in daily driving because of the rapid growth of urban traffic.

5.       What does it mean “A poor CF 32 performance will lead to traffic congestion and traffic oscillation” lines 32-33?  Authors need to explain what they mean.

6.       The sentence “For traditional CF strategies… line 85-87 does not read well, need rewriting.

7.       What is the FV speed distribution? (please define frequency speed at least once) and why is it used?  What other alternatives could have been used?

8.       Sentence on lines 57-59 is not complete.

9.       Line 84 people should be researchers?

10.   Why the Spearman correlation coefficients, justify?  Page 5 line 235.

11.   Table 1:  Something wrong with the table heading (Table 1)

12.   All equations should be numbered and named as equation 1, 2, etc. and should be crossed referenced in the text just before them. 

13.   ALL symbols in ALL equations should be explained!

14.   In this article, a reference to only equation 4 is mentioned although there is no naming for equation 4.

15.   In the discussion and conclusion sections, make a reference or comparisons with previous findings from literature as appropriate.

Reviewer 3 Report

In this manuscript, the authors propose a Deep Reinforcement Learning (DRL)-based autonomous Car-Following (CF) decision-making strategy by Naturalist Driving Data (NDD). However, I will comment on some aspects of the article, and the authors must present the respective changes highlighted:

-The authors incorrectly write the acronyms. The correct form is to write them with the meaning of the acronym with the first initial letter, such as Car-Following (CF). This error must be fixed throughout the manuscript with all acronyms.

-Some acronyms are not with their respective meaning.

-In Figure 1, the unit of speed is m/s.

-Review the units of the measurements written in the manuscript since many of them include a “-”, or they are not represented correctly.

-Equation 1 has not explained each of the terms.

-The title of Table 1 does not correspond to what is presented.

-The authors have not written in the corresponding verbal tense according to the Sections they present.

-The Equations of the manuscript have not been cited, and their terms have not been explained.

-Authors must not use contractions or apostrophes in a scientific article.

-When an object of the scientific article is cited, be it "Equation", "Figure", "Section", "Algorithm", or "Table", it must be done with the complete word without abbreviating and with the first letter in capital letters.

-The authors do not specify the software they used to perform the simulations.

-The authors do not specify or give the parameters that a scenario contains for the experimental simulation.

-The authors do not specify where the data to perform the simulation has been obtained.

-The authors must separate Section 5 into a Simulation Section, a Results Section and a Discussion Section.

-The authors must improve the conclusions they have obtained.

Reviewer 4 Report

The research proposes a car-following model based on deep reinforcement learning. The proposed reward function is designed for safety, efficiency, and comfortness. It seems this is not the first research that model car-following behaviors by considering safety, efficiency, and comfortness, for example, Zhu, M., Wang, Y., Pu, Z., Hu, J., Wang, X., & Ke, R. (2020). Safe, efficient, and comfortable velocity control based on reinforcement learning for autonomous driving. Transportation Research Part C: Emerging Technologies117, 102662.

By reviewing the paper, I did not see the obvious novelty of the research compared with the existing studies. Please thoroughly explain the contribution of the current research.

Round 2

Reviewer 3 Report

The authors have not performed the changes suggested in the first review.

The authors have performed some suggested changes, and have not exactly resolved some questions that have been asked, which this answers must be included in the manuscript.

-There are no changes performed in the acronyms, as must correspond in a scientific paper. -There are Equations that have not yet explained the terms. -The mathematical connotation must be correct to the Equation. -There are acronyms that the authors have not written their meaning. -The authors still do not clarify which simulator they have used, and which is the source of the data that the experiments have developed. -The authors have not separated Section 5 as requested by the reviewers. -The authors have not shown an improvement in the conclusions.

Reviewer 4 Report

1, The authors need to explicitly explain what the advantages of the proposed reward function is compared with the four existing reward functions presented in the response.

2, The authors mentioned the future research includes build personalized driving model using collected driving behavior data. I would suggest that the future research should consider the information collected by both road-side and in-vehicle sensors. They are the complementary information which will facilitate the model training. (please see the following reference as examples).

Hu, J., Zhang, X., & Maybank, S. (2020). Abnormal driving detection with normalized driving behavior data: a deep learning approach. IEEE transactions on vehicular technology69(7), 6943-6951.

Zhuang, Yifan, et al. "Illumination and temperature-aware multispectral networks for edge-computing-enabled pedestrian detection." IEEE Transactions on Network Science and Engineering 9.3 (2021): 1282-1295.

Pu, Z., Cui, Z., Tang, J., Wang, S., & Wang, Y. (2021). Multi-modal traffic speed monitoring: A real-time system based on passive Wi-Fi and bluetooth sensing technology. IEEE Internet of Things Journal.

Round 3

Reviewer 3 Report

Thanks to the authors for performing the suggested changes. Before publishing this manuscript, the authors must perform a minor spell check and correct the acronyms correctly written.

Reviewer 4 Report

Please conduct a round of proofreading to polish the language issues.
